# Community Management and the Demand for 'Water for All' in Angola's *Musseques*

**Allan Cain** * **and Afonso Cupi Baptista**

Development Workshop Angola, Luanda CP3360, Angola; kupydadao@hotmail.com
* Correspondence: allan.devworks@angonet.org; Tel.: +244-912-507253

**Abstract:** The Angolan State's post-war center-piece reconstruction program, to provide the human right to 'Water to All', remains incomplete. The majority of Angola's peri-urban communities still use the informal market to fill the gap. Water selling is the largest sub-sector of Luanda's extensive informal economy, involving extractors, transporters and retailers. Negotiating for water at the local household level involves significant trading in social capital. Communities in Angola's *musseques* have built on neighborhood solidarity to manage the supply of water themselves. The article is drawn from the authors' experience in practice to examine the complexity of Angola's informal water economy and local-level innovative responses. The Government has drawn on these lessons and adopted the community management model MoGeCA (the Portuguese language acronym for Model of Community Water Management)to help address the shortfall. The article is written from a practitioner's point of view, based on more than a decade of experimentation in practice and support from USAID and UNICEF in taking community management to the national scale.

**Keywords:** *musseque*; water-governance; slum; post-conflict; human rights; community management

---

## 1. Context—Angola's Informal Water Economy in War and Peace

The breadth of the 'informal sector' of Angola—and its ubiquitous role in the lives of most Angolans—is evident in Luanda after the war. Throughout the vast slums of peri-urban Luanda where almost half of the city's population currently resides, basic services, such as water and food distribution, are primarily provided by private initiative in the informal sector. The informal economy evolved during the conflict years before 2002, when 37% of the entire country's labor force was employed—likely an underestimate [1]—, robustly justifying the assessment in a UNDP report [2] that "Luanda had become the largest laboratory for survival strategies in the world." In Luanda, strategically located urban markets connected to dozens of smaller satellite markets dominated wholesale and retail trade, acting both as a distribution point for agricultural goods and as the primary source of imported and domestic products for both urban buyers and rural traders [3].

During the war years, the informal sector provided the only potential source of economic opportunities and development for many Angolan households. The current article is based on studies that aim to provide basic information about the structure of the informal market, as well as the knowledge gained from sub-sector studies on peri-urban water market dynamics. The full documentation and analysis of the informal economy in Angola as an integrated whole was required in order to discover how it could be transformed to meet the new challenges and opportunities emerging since the war and how these changes could affect the livelihoods of the millions of Angolans who depended on it. The approach required action-focused research to inform the re-design of development projects to account for new post-conflict realities; to foster practices of good governance and civil society capacity-building in Angola, to empower local communities through participatory approaches, and to enlarge the range of stakeholders who participate in public-policy making. Demonstrating the research

results to policy-makers provides channels through which the perspectives of the poor gain greater voice in public policy-making. Co-production was crucial to this, where community associations, academics, and local government are engaged together in the research and data collection. Co-production also means co-ownership of findings and, consequently, a greater openness by Government to accepting validated data that demonstrates less than favorable indicators, and feeding it into policy discussions. The research presented in this article was collected by teams made up of members of local residents associations, together with technicians from municipal administrations and university students supervised by Development Workshop members.

Probably the largest sub-sector of Luanda's extensive informal economy is water selling in its various forms, and the interface between transporters and retailers is central to the informal water supply. The overwhelming majority of the peri-urban population of Angola depends upon informal systems for their water supplies. Typically, these systems involve buying water from tank owners who purchased their water from lorry owners carrying water from the nearest river. This water is expensive and of poor quality, representing both a significant household expenditure for the urban poor and an increasing health hazard, as evidenced by outbreaks of highly transmissible diseases (such as cholera) known to correlate with poor water quality and limited access (There were over 50,000 cholera cases in 2006 and over 5000 deaths. Cholera and other diarrheas are endemic in Luanda and recur in years of heavy rainfall.). Hampered by the limited capacity of the Government to maintain the already existing rapidly deteriorating infrastructure, the exponential population growth of Angola's major cities has outpaced progress on the plans to upgrade major supply networks to accommodate peri-urban areas [4].

Luanda's water infrastructure, built for a colonial population of half a million in the early 1970s, could not be stretched out to serve the four million people who lived in the capital by the end of the war in 2002. The lack of public investment during the conflict years meant that the informal sector emerged as the main supplier of basic services to most urban and peri-urban families, including water [5].

The subsequent investments in urban infrastructure that were made as part of post-war rehabilitation programs extended the water supply network to serve new middle-class condominiums and enhance the central business district infrastructure. Little was spent, however, in improving the basic services of informal *musseque* settlements. By 2020, Luanda, with a population growth of almost 7% annually, had over 8 million people [6], on track to becoming one of Africa's largest metropolitan regions. Post-war measures to boost water supply in peri-urban areas have focused on installing household connections but failed to adequately maintain the existing network of community standpoints.

In 2007, the Angolan president's program,' Water for All', was initiated to achieve the Millennium Development Goals (MDGs) for water supply. The plan aimed at building new water systems, from groundwater extraction, as well as surface sources (river or lake), to ensure improved drinking water for 80% of peri-urban and rural communities, as well as guarantee a minimum daily intake of 40 liters of water per capita. The proposed infrastructure would include networks for storage, treatment, and distribution, as well as a network of peri-urban supply standpoints. MDG goals were not reached by 2015, and the program's own government assessment found that it did not meet the quality and efficacy objectives and that only 50.3% of the target population was achieved. The fact is that many of the installed systems (mostly diesel-powered generators, submersible pumps, and expensive treatment systems) remain non-functional, and communities therefore still rely on conventional unimproved water sources [5].

During that post-war period, much of the peri-urban population of Angola continued to rely on informal systems for their water supplies. Sometimes lorry-drivers could fill up at official stations (colloquially known as *girafas*) in a few urban districts where treated water comes from the piped supply. In inland cities of Angola, other informal sources of water are available, such as hand-dug wells, improved wells, and boreholes with hand-pumps, but, in Luanda, the water table is too deep to extract ground water economically [5].

While the informal water market continues to respond to the demand for basic services that the State can still not satisfy, the Government sees these vendors as the enemy of modern development, branding them profiteers or "water traffickers". However, the authors argue that the Government should not criminalize the informal market but view the operators as allies [7,8] (interviews on DW's research on urban water markets and equitable basic service fees in Angolan newspapers and radio programs), at least until the capacity of public utilities has been extended to serve all urban citizens. Understanding the informal water supply market was important to see if and how it could be adjusted to function in conjunction with the official supply system, helping to fill the gaps until all urban users were reached. Figure 1 shows the trajectory of the growth of formal water supply systems in urban areas of Angola during the years of war and peace. Improved water systems include household connections but also community standposts and public protected wells that are within 100 meters of households. The graph in Figure 1 ([9] p. 140) demonstrates that about one third on all city dwellers, representing a large majority of *musseque* dwellers, still depend on the informal market for their water.

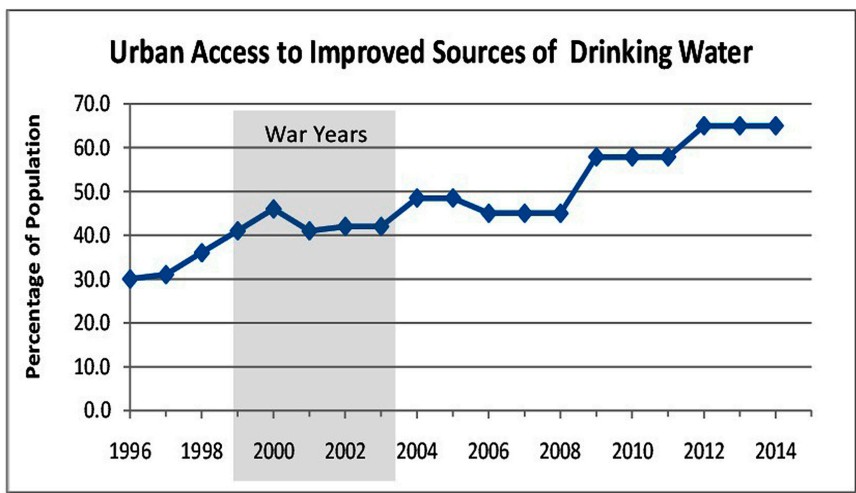

**Figure 1.** Evolution of better potable water supplies in war and peace.

Water supplied by the informal market in Luanda, while partially filling a shortfall left by the State, is insufficient in quantity, expensive, and often contaminated. However, local communities, depending on informal water markets, have shown how their social solidarity has taught them how to develop a water management system in the slums of Luanda. While not fully meeting the human right standard for equitable access to water by all (Sustainable Development Goal 6 aims to ensure availability and sustainable management of water and sanitation for all through universal and equitable access to safe, affordable drinking water.), local communities have shown how building on social solidarity they have found ways to make water more affordable by accepting maintenance responsibility themselves and paying service fees through the collective management of public standposts.

This article is written as a case-study from a practitioner's point of view, based on three decades of practice and innovation. This is not an academic research paper but draws on evidence gathered from participatory diagnostic tools, piloting of practical solutions, the monitoring of this experimentation, and drawing lessons learned and using them to advocate for scaling up with international donors support but, more importantly, for lobbing for changes in government policy that can lead to national-scale replication.

In the mid-1980s, the author's organization, Development Workshop, was invited by the Angolan Women's Organization (OMA) to assemble a team to work with them to address the overwhelming concern to their members, who often spent hours searching and transporting water to their homes every day [10]. Improving access to water was a key to empowering women for OMA and helping free girls from the servitude of carting water, giving them more time to go to school. In the 1990s the World Bank commissioned in-depth studies on water markets in Luanda and other cities as part

of the feasibility-study phase prior to making major infrastructure investments [11,12], aimed at understanding the informal economy's contribution to Angola's post-war development.

An analysis of the principal informal supply chains of the sub-sector was implemented in Luanda and built on and compared with previous research. Geographic Information Systems were used to map the spatial and economic information on an urban-neighborhood, or '*bairro*', basis in a way that was illustrative and understandable to local officials and community leaders [13–15].

Some of the key questions that were asked included: Has the informal water economy changed from the survival mode to a livelihoods development mode in the post-conflict period? How does the informal water economy work? Who are the key actors? What are their relationships? What is the relationship between formal and informal systems and institutions? How does the price for water vary across the city and what are those factors that affect the prices?

The analysis of the water market permitted the development of a better understanding of how informal markets evolved to provide the access to water for communities in informal settlements. The understanding of the sector and evidence from findings were used to feed policy advocacy and promote pro-poor access to better services.

## 2. Water Service Delivery in Angola's Musseques

Water costs were found to vary significantly across districts depending on the type of supply and difficulties related to access. Water costs in Luanda were high due to the huge demand and the distance from the rivers inthe north and south of the city to the surface sources. It has been found that the average household spends over 4% of the household budget on water, and there are a large number of households that spend more than 5%. The very poorest households had to pay 15 to 20%of their household incomes on water [13]. The high cost of water invariably had the effect in many households to reduce a family's consumption or divert funds to other basic needs, such as food and medical expenses. The main determinant of the water price in Luanda is whether or not one lives in the city's urbanized and serviced districts, or in the peri-urban informal settlements '*musseques*' [4].

Many people have to buy water from water vendors in the peri-urban areas, and the price can be as high as 10 times the official rate paid by those who have domestic connections to their homes. Water prices are influenced by the availability or proximity of water from the piped system. Water from a household tank supplied by a lorry is always much higher than in a tank supplied by the piped system (even in the same bairro or street). In general, water prices rise as a function of distance from the piped supply and from the Bengo River (the primary source of water). Difficult road conditions especially in the rainy season, preventing easy access to water trucks often affect the price. The overall average water consumption in Luanda's *musseques* at the time of study was only 22 liters per person per day, which is relatively low but comparable to the other African cities where water is costly and scarce, as well [5,13].

A value or supply chain analysis, illustrated in Figure 2, proved to be an effective tool for understanding the complexity of water markets in Luanda and intersections of both the formal and informal sectors. It was useful to subdivide the market segments into three main components—sourcing, distributing, and consuming—and look at values added and costs attributed at the different stages. The main sources that feed the city, their locations, and the cost of water at the points of origin were identified. The formal and informal chains of water distributors were mapped along with their costs to make deliveries. The consumer is the end-point of the chain and has to pay the accumulated cost of water. Access, satisfaction, affordability, and willingness to pay were all measured. The value chain was used to chart the sequence of transactions where water resources could be unbundled into the different components of the distribution chain, from output to customers and eventual disposal. The services provided in the supply chain encompass all the stages from river extraction to water sales to consumers [5].

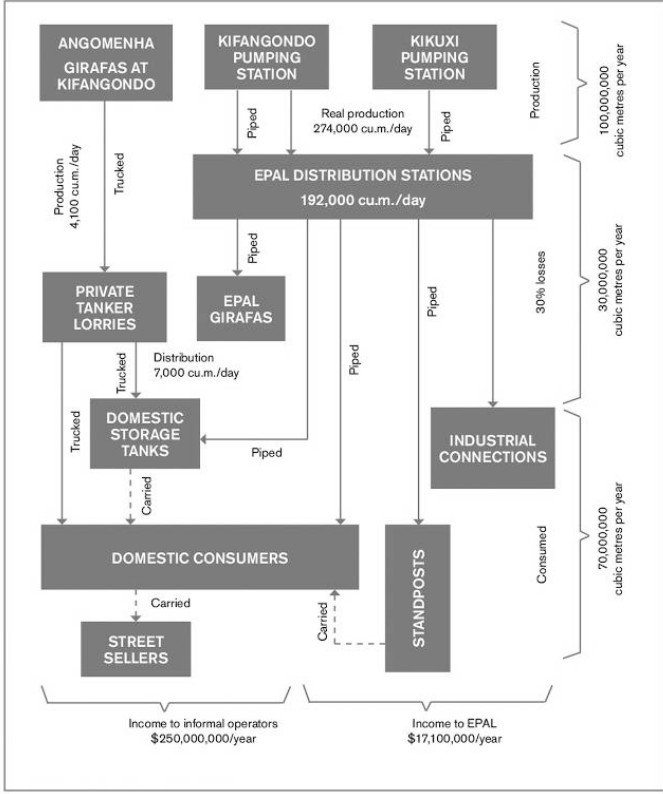

**Figure 2.** Value chain model of Luanda's water supply system.

The market players include: the Government of Angola represented by the National Water Directorate (DNA), which sets policy; the Services Regulation Institute (IRSEA) that sets official prices; EPAL, the public company responsible for water production and distribution in Luanda; and water truck operators and home tank owners, who are key players in the informal sector.

The truck operators take on the task of transporting water to unconnected consumers, while the owners of home tanks provide water from underground tanks for retail sale. Two-thirds of the population in the *musseques* of Luanda is supplied by water truck operators and standposts. The remainder has irregular supply from piped household connections that the Government installed at the time of the Water-for-All Program. Rapid population growth has meant that the proportion of those families with formal piped connections in their own residence has increased only slowly since the end of the war. The standpost system built off of the rudimentary piped network carrying treated water serves a large percentage of unconnected consumers in peri-urban areas. Standposts are considered an interim solution before water pipes can be installed to individual households. End-consumers in the formal piped delivery system on the receiving end of the chain, who are lucky enough to have household connections, are only obliged to pay their water bills to the water utility company and often re-sell some of their water to their unconnected neighbors.

## 2.1. Water Service Providers in the Formal Sector

The informal peri-urban water market in Luanda generated more than USD$250 million per annum. In 2008, it only provided about 20 liters of water per person per day to approximately 4 million people living in peri-urban districts at an average price of about USD$0.01 per liter [16]. The provincial water company is EPAL, the only public sector service provider. If working at full capacity, EPAL could provide 57 liters per person per day for every inhabitant in Luanda. Nevertheless, it supplied just 37 liters per day to the households with domestic connections at the time of the study [16]. Leakage in distribution systems and other technological and managerial issues were major problems.

The amount of water that reaches connected and unconnected consumers cannot be estimated with certainty because there are too many illegal connections to the pipeline in *bairros* without a secondary network distribution system to the households. In addition, intermittent water supply to households with connections means that they also buy water from the informal distributors at various times during the year. *Musseque* households, close to the better served urbanized *bairros*, often access water from clandestine connections to officially served households. Unconnected consumers were obliged to use public standposts, water truck operators, and home tank owners to access water. The majority of the tanks (some 86%) are filled with water supplied by the water truck operators, 11% are connected to a pipeline, and 4% receive water from a combination of trucks and pipeline, depending on availability [4].

As a short-term solution to the water distribution problem, EPAL built water truck filling stations *girafas,* to supply the underserved peri-urban areas of the city, which help serve off-the-network consumers. Operators of water trucks buy treated water at filling stations and then resell it to unconnected households. Most of the filling stations were installed next to the city's EPAL water treatment centers (One of the EPAL sub-systems distributed raw water pumped directly from the Bengo River and treated it in stations within the city.), where the water pressures were higher and where EPAL felt they could be monitored more easily.

Transport distance determines the price of water. Many households are supplied by tanker trucks that carry water from the filling stations that are up to 20 Km away. Water delivery trucks have to circulate in the crowded city streets and deal with the traffic congestion of Luanda.

New water systems serving individual households were built to serve the high-income greenfield commercial areas being built in the city's southern suburb, Luanda Sul. Such new systems also bypass low-income communities with high density on the route. Water infrastructure in the newly developed subdivisions significantly increases the value of land plots for the private benefit of real-estate developers. However, little revenue is earned by EPAL, which sells water at the official subsidized price to a relatively small number of customers residing in these low density housing areas [5]. The official subsidized price for water in 2017 was USD$0.25 [17], compared to the informal market price averaging about USD$10.00 per cubic meter, forty times more. The map in Figure 3 shows the variation of water prices across the city of Luanda, demonstrating the high cost of water in the informal *musseques*.

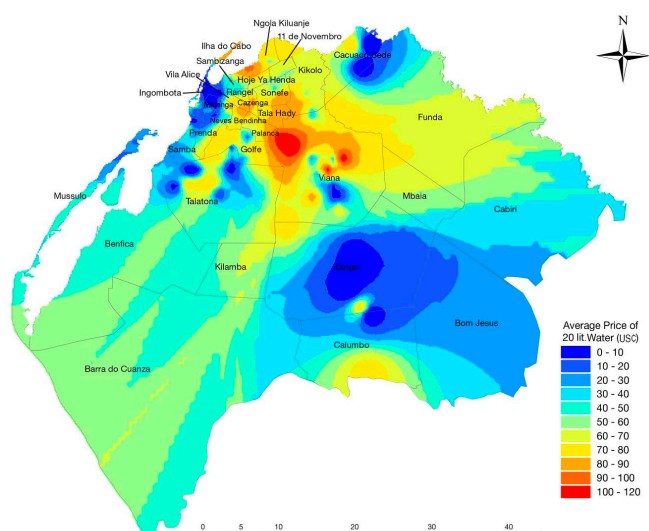

**Figure 3.** Map of Luanda showing the variation of water prices per 20 liter jerrycan.

## 2.2. Informal Providers of Water Services

Unable to meet all of the city's water needs, especially in new informal settlements, the provincial government came to recognize the essential role of water truck operators who have helped in bridging

the gap in the water supply chain. The principal water truck operators supplying the informal water market have registered themselves as an association called ANGOMENHA, which stands for *Associação de Captadores e Transportadores de água de Angola*, the Angolan water pumpers and transporters association (https://www.facebook.com/Angomenha-590504101055066/). The distinction between the formal and the informal is not clearly defined. In the water sector, the informal can often be seen as an extension of the formal—making up for the lack of the formal sector's ability to expand the services beyond the official household connection network. Informal water operators, some of whom are military officers or members of government functionaries' families, prefer to see themselves as government partner-allies, and government officials have often given them due recognition. In reality, they are significant and critical actors in the market chain [5]. They struggle to project their legitimacy as operators with the National Water Directorate, EPAL and the Municipalities.

The principal river-water pumping station is owned by members of ANGOMENHA. This filling station is the main source of water supplied by trucks in Luanda and is located adjacent to the Bengo River in Kifangondo (Cacuaco). The Kifangondo filling station serves around 550 trucks per day, each with a capacity ranging from 5m$^3$ to 25m$^3$.

ANGOMENHA has the most reliable filling system and drivers need not wait in long queues, as seen in Figure 4. The system ensures that water flows continuously each day. (Each pump earns between USD \$100 and USD \$150 per day with trucks paying USD \$0.88/m$^3$ of water. Modest gross revenue for the association can reach close to USD \$5,000, assuming no major breakdowns occur and the official tariff is maintained throughout the month. The pumping station in turn earns annual gross revenue of \$470,000 to \$700,000 a year against running costs for fuel, operator maintenance, and spare-parts.).

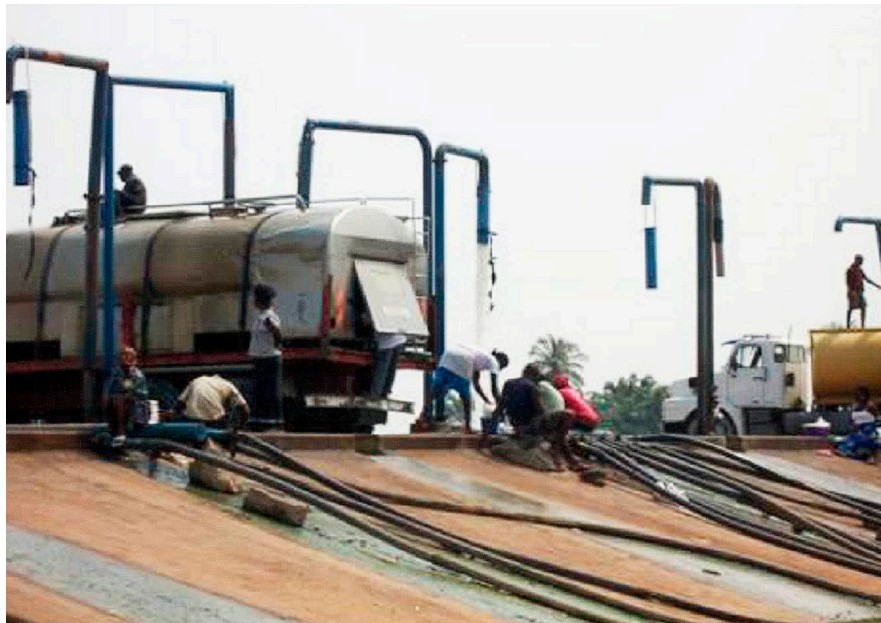

**Figure 4.** Photo of tanker-trucks filling up from Bengo River.

Every ANGOMENHA pump owner member and the water truck operators contribute to monthly system maintenance and a monthly tax payable to the Ministry of Finance of 1%. The members of ANGOMENHA are effectively all individually informal operators. The organization itself is being developed as an effort to formalize and rationalize a key part of the supply chain. The ability of the informal operators to pay taxes and water fees is a proof of their embracing some degree of control. These taxes and other expenses incurred within the supply chain are passed on to the end consumer by water truck operators. Consequently, customers indirectly pay tax for the often poor quality river water they purchase [5].

Water sales are exceptionally profitable for pump operators, who can start earning a profit on their investment in two years or less, but are less lucrative for truck operators who have high driver and fuel labor costs, exacerbated by long, wasted hours spent on overcrowded roads in Luanda. The average income for 10 cubic meter tanker-truck operators is just over USD$900 per week, though the driver's wage, the vehicle's depreciation, and the owner's benefit have to come from this. The most significant cost is the lorry's depreciation rate, calculated to be more than half the cost of running a tanker-truck. The tanker-truck operational cost contributes to a high water price to the customer rather than any unfair profits of the owners [5].

The water sold at the filling station of ANGOMENHA is untreated at the source, posing potentially serious health risks. The association, however, did provide for water treatment. All drivers are expected to stop in a small water treatment station nearby for water chlorination. While chlorine treatment costs just USD$0.12/m$^3$, there is no mechanism for forcing truck drivers to stop or toensure that the water has been decontaminated successfully, and EPAL, the provincial water authority, takes weekly samples of water, which are taken to their labs for chlorine analysis, but only from the cisterns of trucks that have voluntarily stopped for chlorination [4].

*2.3. Informal Home-Based Water Retailers*

The link between the transporters and the seller is a critical point in the informal water supply chain. This is where prices are determined and the amount of water available to the household-based re-seller is determined. Water truck operators bring water from the main filling stations to unconnected household retailers. The retailers, in turn, service an estimated 70% of the peri-urban population of Luanda, who are either not connected directly or are indirectly connected to the formal network through standposts [5].

The wholesale price is negotiated between the household-based reseller and the trucker and will depend on distance and other supply and demand factors. The water purchased is stored in householders' yard tanks that are usually underground. The majority of the tanks (some 86%) are filled with water supplied by the water truck operators, 11% are connected to a pipeline, and 4% receive water from a combination of trucks and pipeline—depending on availability.

Water is purchased for both family consumption and neighborhood re-sale, as seen in Figure 5. The underground tanks are made of concrete blocks with a storage capacity of 5 m$^3$ to 15 m$^3$. Water retail prices are set by home-based resellers. When they can buy bulk water cheaply, these savings are normally passed on to consumer neighbors. Vendors seldom sell for profit but instead cover their own cost of water consumption [4]. When interviewed, their neighbors said that they rarely felt that they were being exploited by the water vendors [11,18].

Water availability and its cost in the neighborhood are not decided by commercial considerations alone. Social relations and solidarity with neighbors play an important role. Social solidarity is complex in cities in post-conflict Angola where most peri-urban residents are former internally displaced persons. Urban populations tend to be heterogeneous with a mixture of old and new settlers, often from different ethnic groups [19] (Robson and Roque, 2001). Neighborhood *bairro* relationships are built on closeness, common local issues, and shared privation. Householders with a water tank can choose not only the amount but also which neighbors they want to sell to. Water prices also differ, depending on the relationship between tank owner and water buyer. The owners of water tanks often sell water for a lower price to people with whom they have, or have built, a relationship or mutual solidarity [18].

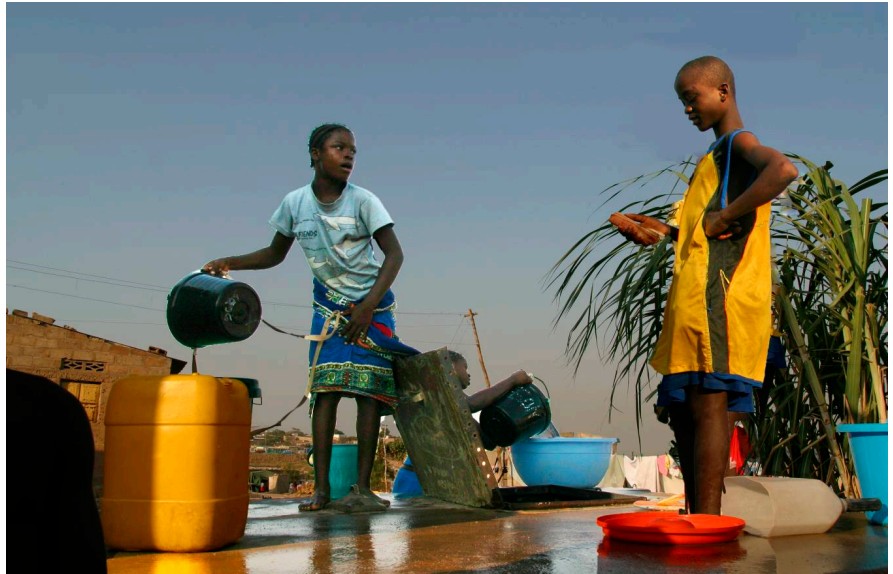

**Figure 5.** Photo of water being resold to neighbors from underground household tanks.

Home tank owners often do not have enough money on-hand to buy a full tanker-truck of water on a regular basis once their own tank is dry. Until they can collect a lump sum to purchase a complete load of water, they can become users of water from other neighborhood tank owners. Social networks evolve locally between neighbors, who may at different times be buyers and sellers. It becomes therefore essential for every water consumer in a poor, un-serviced *bairro* to maintain friendly social relations with a range of water suppliers within walking distance of their homes.

Home water retailers typically did not develop networks outside their neighborhoods. Despite public health campaigns, aimed at building awareness of the dangers of consuming un-treated water, domestic water-tank owners have not built associations like ANGOMENHA. There are no seller networks that have demonstrated capacity to leverage tanker truck operators to better control the quality of water or guarantee its pre-treatment. Public-health authorities therefore have had to promote household water treatment and improved hygienic storage through radio campaigns and social media.

### 2.4. Water Street-Sellers

Street vendors, as seen in Figure 6, who work in the informal market selling water in small containers or plastic bags, also perform a secondary level of retailing. Usually these vendors receive their water from home tanks and standposts and sell in half liter units for the equivalent of USD$0.05 to 0.10. Water sale on the street and in markets is often performed by ambulant traders, who are usually considered to be at one of the water market's lowest rungs and make only marginal incomes.

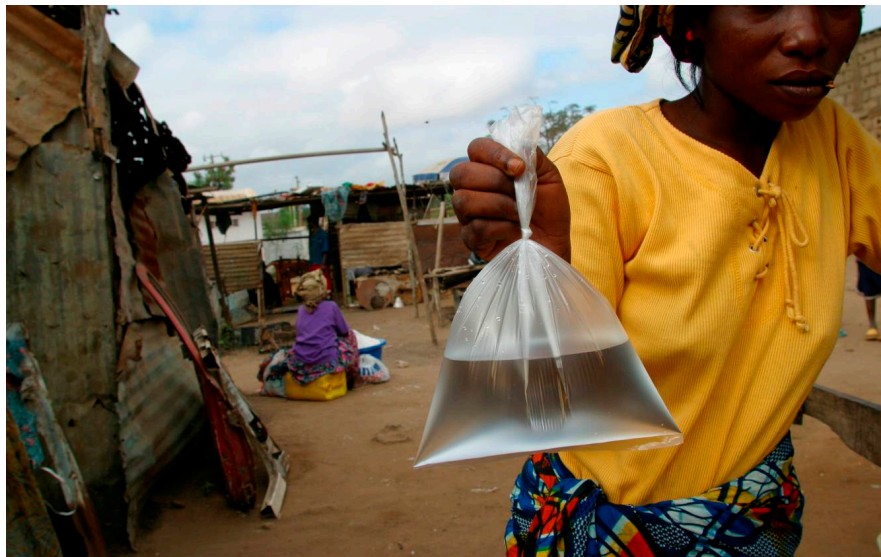

**Figure 6.** Photo of street vendors reselling water in plastic bags in the informal market.

### 2.5. Water Carriers and Stevedores

Water transport within neighborhoods by women and girls, who account for 85% of carriers, is rarely factored into the price of water after it is delivered by truck to the owner-reseller of the neighborhood tank or by pipe to the standpost. Women and girls who carry jerrycans, basins, or buckets, sometimes hundreds of meters, to their homes add significant time and therefore value. Child stevedores, seen in Figures 7 and 8, are hired to move water carts, typically weighing between 40 and 50 kilograms, for longer distances of up to several kilometers.

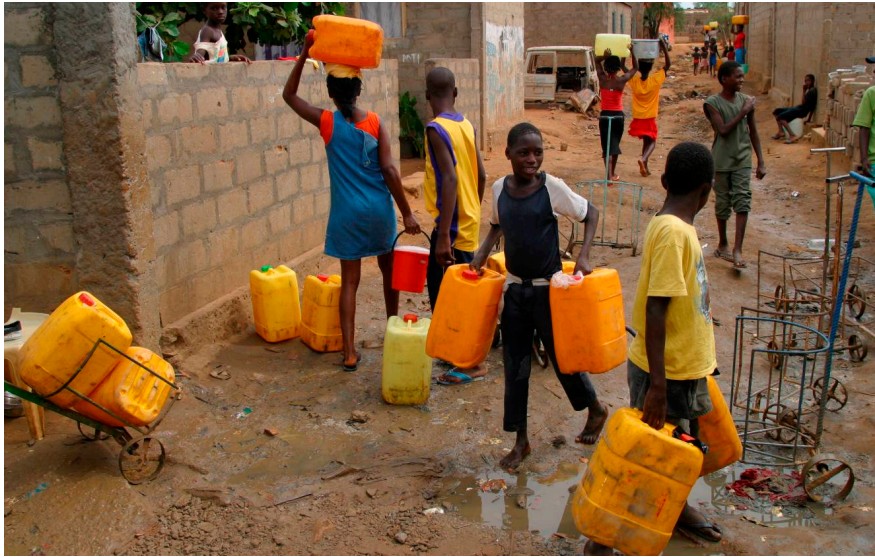

**Figure 7.** Labor of women and children is rarely factored into the price of water.

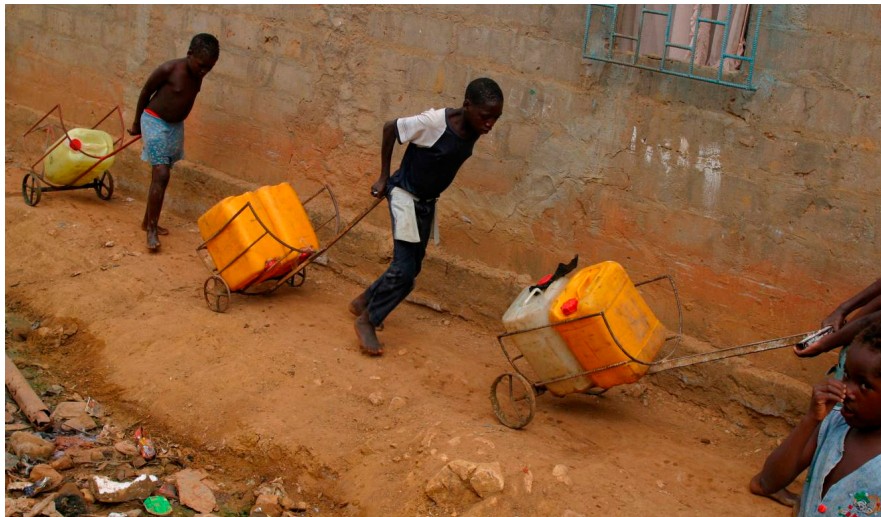

**Figure 8.** Children hauling water carts often weighing 40–50 kg.

## 3. Challenges Related to the Human Right to Water

Within international forums, the Angolan Government has accepted its responsibility as *duty-bearer* to deliver sufficient, safe, and affordable water and sanitation, equitably, to all citizens and has attempted to bring its national policies into conformity with the Millennium and later the Sustainable Development Goals. Despite the demand of citizens for their right to water, Angolan cities are likely to continue depending, for some time, on informal water suppliers. The Government has tended to regard informal providers as economic opportunists, or worse as black-marketeers and exploiters of the poor, because of the high costs of transport which truckers are obliged to pass on to consumers. Critics claim that small-scale informal water service providers have poor customer service and fail to meet both technical and quality standards. However, because the State's formal water services sector has been unable to meet the water needs of so many residents in Luanda, officials have been forced to accept the role of informal water truck operators who fill the supply gap while the Government is developing its capacity to fulfill its full public obligations [4].

### 3.1. Water Policy Reform

Substantive water sector reform was introduced with the approval of the National Water Law in 2002, shortly after the signing of the peace accords. The new Law included key principals: an acknowledgement of water as both an economic and social good; the acceptance of alternative models for urban service provision that should take place at the lowest possible level; the institution of autonomous water utilities; and the development of a comprehensive tariff strategy. Reform of the water sector was necessary in order to attract the major investments that were eventually provided by the World Bank, the European Union, and the African Development Bank. The Angolan Government also used commercial credit lines, such as those now accessible from China and Brazil, for financing water infrastructure [5].

Richer neighborhoods where household connections to the water supply network had long benefited from subsidized water prices. The Government's new policy (The collapse of global oil prices in 2014 obliged the Angolan Government to follow recommendations from the World Bank and IMF to progressively remove subsidies from basic services and fuel.) of dismantling subsidies and implementing a cost recovery framework for basic services across-the-board, addressed, to an extent, the demand for a more equitable approach to rights and responsibilities.

The 'Water for All Program' became Angola's showcase program for the development of drinking water. Launched in 2007, it was designed to run until 2012. The program built or rehabilitated 638 water sources, consisting mainly of community water points and hand pumps, and installed or restored

282 small supply systems, which were made with electrical pumps and elevated water reservoirs. The program achieved coverage of over 50%, although it did not meet its ambitious goal of reaching 80% by 2012.

*3.2. Water for All?*

A 2012 study commissioned by the African Development Bank (AFDB) for the National Water Directorate (DNA) (Development Workshop and Cowater International implemented the study for AFDB between 2013 and 2015.) shows that in many communities, the Government's flagship 'Water for All Program' has failed to ensure effective access to water on a sustainable basis. Beneficiary communities were rarely involved in the evaluation of needs and their desire for a water system, as well as its opinion on the choice of technology and management model. Of the electric pump and water treatment systems installed, only 48% were functional. The main constraint identified during the design of the program was the availability of public utility implementation and private sector capacity. This study also found that local government capacity for water systems operation and maintenance is very low. There was almost no collection of user fees and local governments and municipalities had a limited sense of local ownership of water systems and therefore felt no responsibility to maintain them. One main conclusion of the study was that these technical solutions were seldom discussed with the communities and usually did not respond to user preferences [5].

Mindful of the shortcomings of the 'Water for All Program', the Ministry of Water and Energy (MINEA) took a decision in 2014 to tackle the sustainability problem by implementing the MoGeCA (Community Water Management Model) as part of Angola's water policy reform, rolling out the program in several provinces. USAID provides financing to Development Workshop to pilot MoGeCA in Huambo, Cunene, Cuanza Sul, and Luanda. The aim of this program was to ensure affordability of water for consumers, while providing for the maintenance of the water infrastructure.

## 4. A Model for Managing Water Sustainably

*4.1. Self-Management-Building Community Social Capital*

During the war years, attempts were made for the supply of basic services, principally water to thousands of displaced families who were fleeing rural conflict zones to the relatively safe havens of cities. Water systems based on wells, hand-pumps, and standposts were constructed with humanitarian assistance to provide emergency supplies to communities of displaced people. Water-point caretakers, as seen in Figure 9, were chosen by each community and trained by humanitarian organizations to carry out basic maintenance and to take over the management of these systems to ensure that water was distributed equitably.

During the protracted years of the war, community self-management of local water supplies had proven to be an effective way of avoiding vandalism, neglect and protecting infrastructure in the interests of displaced communities. Even in the post-war period, when international humanitarian aid was withdrawn, the continued limited capacity of public utilities providers and local authorities meant that management strategies, which included consumers in the delivery and maintenance of basic services, needed to be established. Through affordability and willingness to pay appraisals, it became apparent that low-income households were prepared to pay for a public water supply if they provided reliable service, and the price was less than that charged by private vendors [11,12,20]. Local residents generally assume that piped water from standpoints on the public grid has been treated, and the standard is superior to the water sold on the informal market by the bucket [4].

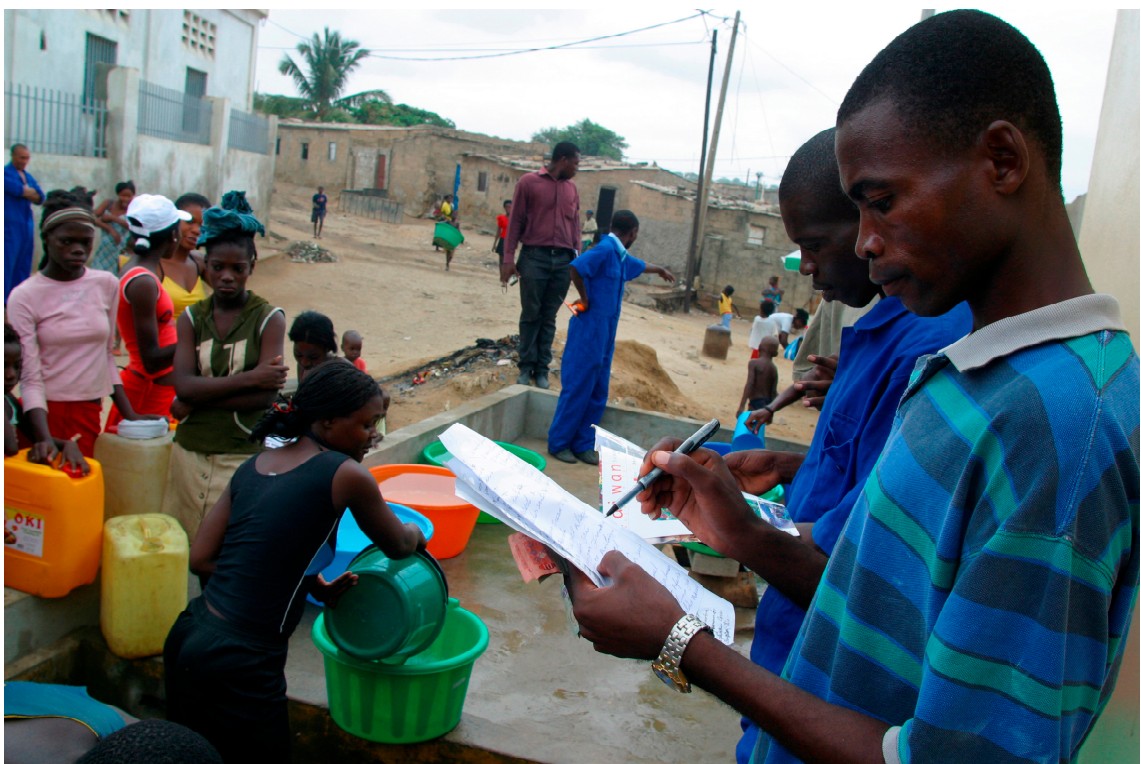

**Figure 9.** Water-point care-takers were elected by each community to take over the management.

*4.2. Community Water Management Through MoGeCA*

Based on the experience of building self-sustaining water systems in the war and post-conflict years a model therefore needed to be promoted to provide an expanded service at an affordable price to consumers. The sustainable community water management model, MoGeCA (the Portuguese language acronym for Model of Community Water Management.), was developed through rigorous practical testing of its components, allowing for sufficient time for learning and feedback. This kind of long-term, local institutional testing is not usually supported by most foreign funding agencies. A support for capital investments in time-bound programs is more frequent. International agencies, such as the World Bank, often promote privatization as a solution [21], but the private sector has shown little interest to enter the sector in Angola [22]. Angola's national private sector is weak, its public institutions are not strong enough to regulate privatization, and Angola's population has incomes too low to be attractive to private enterprises.

The Luanda water company EPAL recognized that while it did not have the capacity to manage water delivery at community level, that it should prioritize the supply of bulk water—that is, improving the process of extracting water from the river, treating it, and distributing it through the municipal networks. Urban district associations of water committees based in *Comunas* were established, through which they could share their experiences and work together to seek better services from EPAL and the municipal authorities [4]. Some standposts date back to before independence in 1975, but most were built by EPAL and Development Workshop with financing from international institutions and the government's Water-for-All Program.

Water committees are eventually to be set up to work at Luanda's 1500 water standposts, to collect revenue, oversee operations and maintenance, track and document the number of water flow days, and ensure records of all payments and expenditures were kept. Water and water-point maintenance costs were recovered from consumers, and EPAL's proceeds allowed it to provide the community water-standposts with a continuous supply. It meant creating community organizations for which users were accountable, something that had no precedent [4].

The traditional two tap open-air standpost was transformed into larger community water kiosks, as shown in Figure 10. Kiosks have six to eight taps and are roofed and protected with grating walls, a lockable gate, and a soak-away drainage system for overflow water. Clothes wash-stands are often installed under the same roof, as seen in Figure 11.

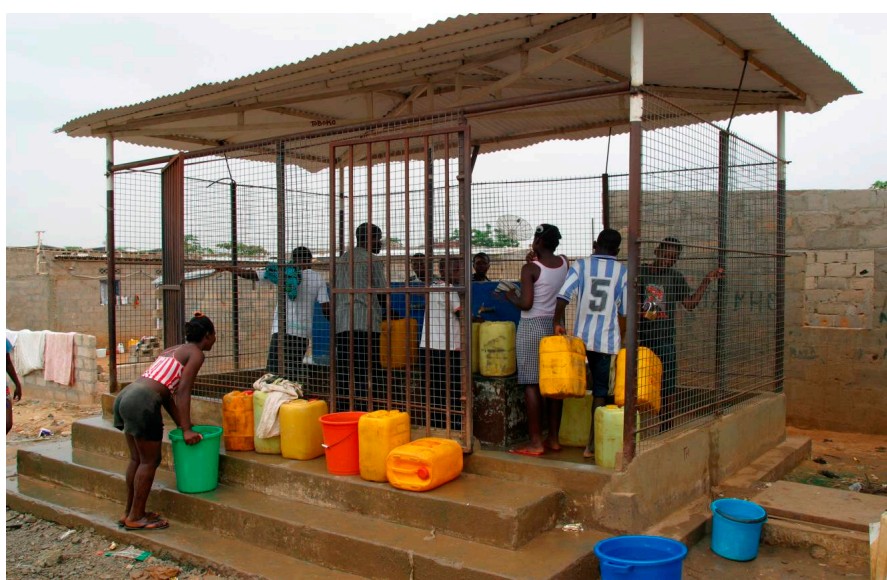

**Figure 10.** Water kiosks, having six to eight taps, have replaced the stand-pipe mode.

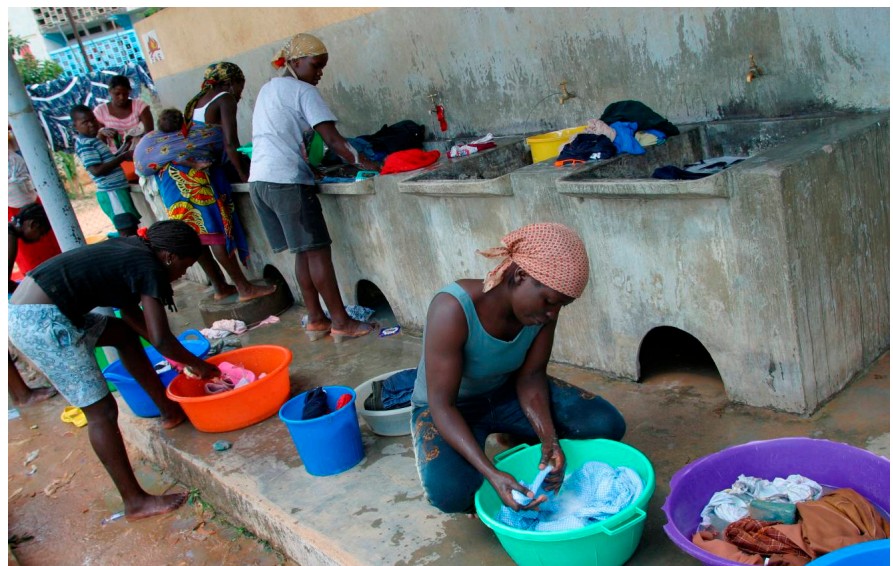

**Figure 11.** Clothes wash-stands are often installed at the kiosks.

Two hundred committees for water kiosks representing more than one hundred thousand consumers had already been formed by 2018. Each committee managed its own finances and handled conflicts, including enforcing the prohibition on illegal connections. By 2018, seventeen district-based associations had been formed, each representing ten to twenty neighborhood water committees but still covering less than ten per cent of Luanda's standpost water consumers.

Local consumer group associations provide the legal support needed to register with municipal governments, and trains community residents to help them reach their potential as water services user managers. In this water supply model, users are client-consumers. They make fair payments at an average of USD$0.05 per 20 liter jerry can, for the services provided. The money collected

is split proportionally to pay for water from EPAL, create a savings fund for the purchase of spare parts, investment in improvements to the standposts, and to support the association. Maintenance funds are managed by the associations by annually audited bank accounts to ensure the quality and accountability of group money management. This strategy helps to guarantee the financial sustainability of the standposts, helps people to become accustomed to paying for public services, and strengthens the capacity of local structures in management and accountability [4].

*4.3. Co-ownership & the Demand for the Right to Water*

Collective ownership is the foundation of the viability of MoGeCA. If the community feels it has a share of ownership of the water infrastructure, it will use the system more wisely and take initiatives to maintain and repair it. Consumers with a sense of entitlement are placing more demands on service providers to improve their quality and efficiency. Economic sustainability is supported by customers' willingness to pay fees for this service [4].

Consumer organizations have shown their ability to monitor reliability and quality of services. By advocating for their "right to water", they have become powerful voices and have been active by ensuring access to better facilities. In the MoGeCA model, it is the community association (ACA) that has responsibility to liaise with the water utility provider (such as EPAL in Luanda). In the case of system failures, however, the communication line proved inefficient and response times were slow. For the system to meet the expectations of the community and respond to emergencies when they occur, a real-time complaint system was needed. Working with the International Association of Mobile Operators [23] and the African Innovations Foundation, a mobile-phone based water monitoring system was piloted called VerAgua, engaging the care-takers at the 200 water points and kiosks (VerAgua is also being piloted in 150 rural water systems in the province of Huambo serving about 100,000 people.) who had already been trained in basic book-keeping and maintenance. Figure 12 shows a water kiosk managed using the cell phone monitoring system.

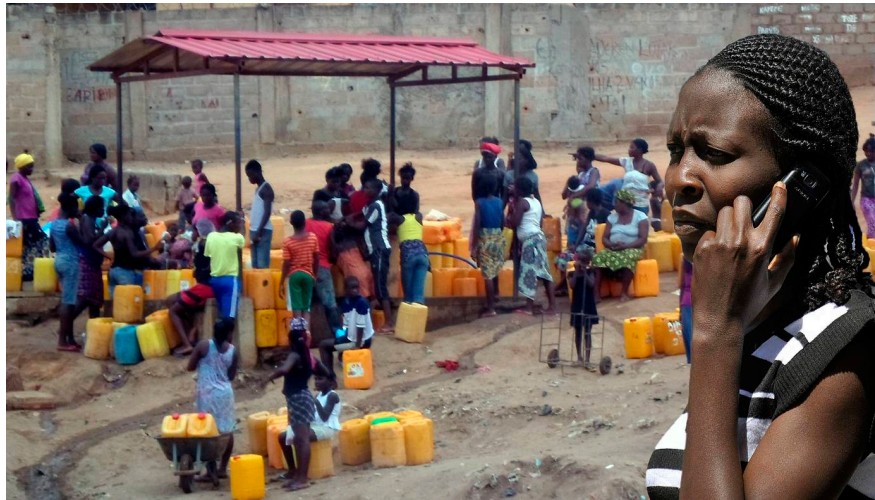

**Figure 12.** Community water management using the mobile-phone-based VerAgua monitoring system.

The VerAgua program uses mobile services (specifically free missed calls, SMS, and data) to relay water-service status information. Mobile phone service providers in Angola do not bill for missed calls which are not picked up. Caretakers of the project have been issued unique telephone numbers that correspond to their specific water standpost location. The caretakers' own phone numbers were also connected to their water point in the database. This allowed the computer-server linked to the GSMA mobile network to connect the missed calls to different water points and to contact the caretakers assigned to each point by Development Workshop. Caretakers reported by placing a missed call to specific phone numbers that match the precise status: related to the functionality of the water

point (functional, partially functional, or non-functional). An SMS was automatically sent back to caretakers within 20 minutes acknowledging confirmed water status. These calls were monitored by the web platform to obtain the status of all water points. A status report was sent to the water supply authority once a week, indicating water point status and information on the operation of standposts. The pilot aims to use water services information to raise awareness of government and build consumer awareness, thereby driving the water company to improve services. Early results are seen as promising, with consumers and service providers alike remaining enthusiastic about expectations that close-to-real-time water monitoring will result in a more sustainable and efficient service [4].

## 5. Conclusions— Can community management deliver the 'right to water' in Angola?

The 'Water for All' program has been only partially successful in addressing the Angolan population's demand for the right to water. However, at least a third of urban dwellers still depend on the informal market and many of the others have had to depend on community solidarity and local action to get access to standposts and communal wells and bore-holes. The Government failed to invest a sufficient portion of their post-war wealth to respond to the priority need of delivering water to at least meet their commitments to the MDGs and SDGs.

Academics and policy-critics in many developing countries debate issues around the community management of water [24]. Those promoting the model argue that community management leads to improved performance, as local technicians are able to respond more quickly to breakdowns and users have a direct interest in making financial contributions through fees to ensure their water supply continues to function [25]. Nevertheless, model critics point out that, by shifting responsibility for maintaining water systems to communities, local and national governments abdicate their long-term responsibility for the delivery of service. International donors support the approach, which allows them to invest in relatively short-term projects that can be completed and then 'handed over' to communities who are expected to sustain the services [26]. A review of the literature on community management concludes that the payment of water user fees is not only essential for sustainability but also leads to the more careful use of water [24].

Ducrot and Bourblanc [27] demonstrate the importance of building social capital in post-conflict Mozambique. They argue that community water interventions must build a sense transparency in access and the fairness in the distribution of benefits. The lack of or equitable access, due to long distances that some poor families must travel, can weaken the buy-in from those less-well served and discourage collective mobilization necessary to maintain community water infrastructure. Inequitable benefits can undermine the sustainability of the systems. Their evidence shows that communities rejected interventions that favored some of families over others. They highlighted the critical role of elected leaders and the danger of those who are self-appointed or customary chiefs who may take on the function of 'gate-keepers' and assuming the control of access to benefits from external interventions, reducing water committees' role to collecting water fees and carrying out minor repairs.

More than 50% of urban population in Sub-Saharan Africa depends on informal, small-scale providers and community initiatives for their drinking water [28]. Mapunda, Chen, and Yu [29], in Tanzania, show that, like in Angola, various types and sizes of informal water suppliers are filling the gap left by public utilities or failed attempts at privatization. They argue that the integration of informal operators into the public regulatory system can greatly contribute to the improvement of water supply services for peri-urban populations.

In Luanda, in 2019, twenty district-based water community associations (ACAs), representing tens of thousands of customers, met at the Annual Community Water Conference. Since external donor support had ceased, each ACA has been self-managed and self-financed for more than a decade. Detailed financial statements were delivered to the members at the annual meeting showing the revenue collected in fees at the standpoints and the funds ultimately forwarded to the water supply

company EPAL for the procurement of bulk water, while 30% was retained for maintenance costs and caretaker subsidies.

Throughout Luanda and other urban centers in Angola, the informal water markets still operate with the supply chains that were described in this paper. Although the water market continues to be considered parallel, it is progressively crossing over and connecting at strategic points with the formal supply system demonstrated in Figure 2. Despite patronage by the State, Angola's private sector was slow in creating and offering sustainable business models. Innovation in the markets for water supply has largely come from the informal sector. This is evidenced by the group of water-truck operators who founded ANGOMENHA to fill a gap in the water system and reach communities beyond the piped water system. Their business model is sustainable because, along the supply chain, the small profits are distributed and they pay their taxes, essentially entering the formal market. The delivery of water by lorry, however, is very expensive and wasteful in energy and labor, and these high costs are passed on to consumers. Even MoGeCA's peri-urban beneficiaries are burdened with paying close to the real cost of water production, while the economic elites, lucky enough to have running water from the piped network in their homes, have used their political weight to actively resist and postpone enforcing government plans to enforce full cost recovery for municipal services [4].

State investment in initiatives, such as 'Water for All', should continue but be phased in, in such a way that *musseque* populations at least have access through standpoints, before individual household connections are built in middle-class neighborhoods. Research done by Development Workshop shows that the low-income individuals, who normally paid high prices for poor quality water, are the most reliable consumer-clients and the most willing to pay for the services.

Poor neighborhoods have shown that social capital generated by participation in water committees can be used in new municipal government councils to gain recognition and give them a platform to voice their demands for more equitable services. Traditional collaboration in accessing scarce water has employed a micro-enterprise approach to neighborhood commerce [4]. MoGeCA has expanded to provide and retain services for an increasing number of communities in Angola on a financially sustainable basis. The Angolan government is promoting the model as a transitional approach to be applied nationwide until all families can be provided with their human right to have access to clean and affordable water in their own households.

**Author Contributions:** A.C. was responsible for the conceptualization and writing of the original draft of this article and the review and editing of the published version; A.C.B. was responsible for the investigation, research and project administration and supervision of the Model of Community Water Management discussed in this article. Both authors have read and agreed to the published version of the manuscript.

**Funding:** This research was funded by USAID, Cooperative Agreement number: AID-654-A-14-00001.

**Acknowledgments:** Development Workshop's research through the war and post-conflict years has been supported by the International Development Research Center, the World Bank, Bill & Melinda Gates Foundation, and the Department for International Development UK. Photographs in Figures 5–11 are provided with permission of Tim Hetherington under commission from Development Workshop.

**Conflicts of Interest:** The authors declare no conflict of interest. The funders had no role in the design of the study; in the collection, analyses, or interpretation of data; in the writing of the manuscript, or in the decision to publish the results.

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
