# Peer review of "Community Management and the Demand for ‘Water for All’ in Angola’s Musseques"

_water, doi:10.3390/w12061592_

Round 1
Reviewer 1 Report
I am delighted to read this paper and to be able to comment, particularly because it focuses on a country we know very little about (Angola) and urban water management (another relatively under-reported country). I also enjoyed the fact that it was written from a practitioner's viewpoint. Water the journal will benefit from having more articles like this one.
I have two minor concerns with the paper. The first one is that it does not discuss a research question. What is the puzzle that this author intends to answer? What is the underlying research question? This paper DOES have a core goal, and the author CAN make it work to answer a research puzzle. He/she only needs to think hard about which question they're attempting to answer.
The second minor concern is obviously the research design. I would simply get rid of "materials and methods, etc." sections and simply just outline learnings and challenges to robust water management. I recommend that the author considers learnings and challenges as the core sections and find ways to bridge them towards the research question.
Author Response
Dear Reviewer
Thank you for your comments to our paper. The revisions that you request suggest transforming the article into an academic research paper format. I am not sure that you were informed that our paper is a response to the call for practitioners and policy-makers to present case studies of their work for the Special Issues on the "Human Rights to Water and Sanitation".
I understand that you are accustomed to contributing to the MDPI's academic research journals. However in referring to the call for proposals, I quote here "We are looking for case studies that reflect the multitude of approaches and ideas across regions and between stakeholders and sectors, and we particularly invite practitioners and decision-makers to share their successes and failures."
The case-study that we presented in this article represents several decades of our work as practitioners in Angola. We are reluctant to re-structure this case study into an academic-research format and fear we would lose the objective of the "Special Issue". We welcome any further editorial comments that you may have on improving the presentation of our case-study.
Best regards
Allan Cain & Afonso Cupi Baptista,
Development Workshop Angola
Reviewer 2 Report
This is more like a review of the substantial work going on in Angola as part of the "Water for all" program. I liked reading the contents and the detailed description of what existed and what is going on with regard to access of water. However, the papers lacks in originality in the sense of having an author engineered model (at least a flow chart to better understand how the results may be used for policy application elsewhere. It reads more like a technical report,so i would encourage the author to summarize and put more thoughts on how the measures undertaken like in the community based projects can be successful. Also, delineate the policies in other African nations and make a comparison if possible.
Author Response
Dear Reviewer
Thank you for your comments on our paper. The revisions that you request suggest transforming the article into an academic research paper format. I am not sure that you were informed that our paper is a response to the call for practitioners and policy-makers to present case studies of their work for the "Special Issue on the Human Rights to Water and Sanitation".
I understand that you may be be accustomed to contributing to the MDPI's academic research journals. However in referring to the call for proposals from the guest editor, Dr. Leo Heller, the UN Special Rapporteur on Human Rights to Water, for this Special Issue, I quote here; "We are looking for case studies that reflect the multitude of approaches and ideas across regions and between stakeholders and sectors, and we particularly invite practitioners and decision makers to share their successes and failures."
The case study that we presented in this article represents several decades of our work as practitioners in Angola. We are reluctant to re-structure this case study into an academic-research format and fear that we would lose the objective of the "Special Issue". We welcome any further editorial comments that you may have on improving the presentation of our case study.
Thanks and best regards
Allan Cain & Afonso Cupi Baptista
Development Workshop Angola
Round 2
Reviewer 2 Report
The paper could be modified with some more comparative analysis with other countries with similar levels of development as Angola. Also, the conclusions need to be strengthened a bit more with how the "Water for all" program may have been partially successful with the informal market dominant and still feeding into the formal sector for allocating water in Angola.
Author Response
Dear Reviewer
Thanks for your second round of comments.
I have done a final English language check and standardised spelling and removed the final "Track Changes".
In response to your comment regarding Research Design and Research Results, I have explained our approach to the article as a case-study drawing on evidence gathered from practice with the aim of influencing public the policy in Angola.
Thanks as well for your recommendation on strengthening the paper with more comparative analysis with other countries with similar levels of development to Angola. I have drawn on case study material from Mozambique and Tanzania, demonstrating that our findings were mirrored in these countries and policy recommendations could be applied more widely in Sub-Saharan Africa where more than 50% of people depend on informal and community access to water services.
I am sending a copy of the article showing additions in red text.
We very much appreciate your advice on improving our article.
Regards
Allan Cain & Afonso Cupi Baptista
